# Triterpenes of *Prunella vulgaris* Inhibit Triple-Negative Breast Cancer by Regulating PTP1B/PI3K/AKT/mTOR and IL-24/CXCL12/CXCR4 Pathways

**DOI:** 10.3390/ijms26051959

**Published:** 2025-02-24

**Authors:** Yamei Li, Hongshan Luo, Xiulian Lin, Linye Hua, Jiayao Wang, Jingchen Xie, Zhimin Zhang, Zhe Shi, Minjie Li, Qiuxian Peng, Limei Lin, Duanfang Liao, Bohou Xia

**Affiliations:** Key Laboratory for Quality Evaluation of Bulk Herbs of Hunan Province, School of Pharmacy, Hunan University of Chinese Medicine, Changsha 410208, China; yameili@hnucm.edu.cn (Y.L.); luohongshanxc@163.com (H.L.); 20232047@stu.hnucm.edu.cn (X.L.); linyehua01@stu.hnucm.edu.cn (L.H.); 11yyxf@stu.hnucm.edu.cn (J.W.); 004108@hnucm.edu.cn (J.X.); cslgdxzzm@163.com (Z.Z.); zhe.shi@hnucm.edu.cn (Z.S.); tfxiaobei@163.com (M.L.); cspqx@163.com (Q.P.); lizasmile@163.com (L.L.)

**Keywords:** triterpenes of *Prunella vulgaris* (PVT), triple-negative breast cancer, PTP1B/PI3K/AKT/mTOR pathway, IL-24/CXCL12/CXCR4 signaling axis

## Abstract

Triple-negative breast cancer (TNBC) is a type of breast cancer characterized by high molecular heterogeneity. Owing to the lack of effective therapeutic strategies, patients with TNBC have a poor prognosis. *Prunella vulgaris* L. has the effects of reducing swelling, dissolving knots and treating breast carbuncles and mammary rocks. Modern pharmacological studies have reported that it can effectively inhibit the growth of breast cancer. The main active antitumor components of *Prunella vulgaris* are triterpenoids (PVT); however, the role and potential mechanism of PVT in TNBC remain unexplored. Our study aimed to further explore the inhibitory effects of PVT on TNBC and the associated mechanism. The results showed that 19 compounds associated with PVT were identified, 9 of which were triterpenoids. The percentages of ursolic acid and oleanolic acid in PVT were 34.51% and 11.32%, respectively. Triterpenes of *Prunella vulgaris* significantly inhibited the proliferation, migration and invasion of MDA-MB-231 cells and promoted their apoptosis in a concentration-dependent manner. PVT could also effectively downregulate the mRNA and protein expression levels of *Ptp1b*, *Pi3k*, *Akt* and *mtor* and upregulate the mRNA and protein expression levels of *Il-24* in MDA-MB-231 cells. In mice with tumors of TNBC, PVT significantly reduced tumor growth and the expression levels of PTP1B, CXCL12, CXCR4, PI3K, AKT, mTOR and other proteins in TNBC tumor tissue and upregulated the expression of IL-24. This study showed that PVT played an anti-TNBC role by regulating the PTP1B/PI3K/AKT/mTOR signaling pathway and the IL-24/CXCL12/CXCR4 signaling axis.

## 1. Introduction

*Prunella vulgaris* L., a species of the *Prunella* genus in the Lamiaceae family, is bitter–pungent and cold in nature, and belongs to the liver and gallbladder meridians [1]. According to ancient books, *Prunella vulgaris* has the effects of reducing swelling, dissolving knots and treating breast carbuncles and mammary rocks. Modern pharmacological studies have reported that *Prunella vulgaris* can effectively inhibit the growth of breast cancer [2].

According to the latest global cancer data reported by the International Agency for Research on Cancer (IARC) of the World Health Organization (WHO), there were an estimated 20 million new cancer cases in the world in 2022, among which breast cancer (BC) had the second highest incidence, accounting for 11.6% [3]. Breast cancer is the most frequently diagnosed cancer and the main cause of death in female cancer patients, and seriously threatens the physical and mental health of women [4]. According to the latest national cancer data issued by China’s National Cancer Center, breast cancer is still the malignant tumor with the highest incidence of female cancer in China, with 357,200 new cases of breast cancer in China in 2022, accounting for 15.4% of the global incidence of breast cancer [5]. Although research on breast cancer has been ongoing for more than 30 years, the treatment of early breast cancer has achieved great progress [6,7,8,9], but the survival rate of patients with triple-negative breast cancer (TNBC), the most dangerous subtype of breast cancer, has not significantly improved, and one of the main reasons for its high mortality and easy recurrence is the invasion and metastasis of cancer cells [10,11]. Triple-negative breast cancer lacks a clear therapeutic target, and it is difficult to achieve benefits from endocrine therapy and targeted therapy, resulting in poor prognosis, high recurrence and metastasis rates and high mortality [12]. At present, the main treatment strategies are surgery, radiotherapy and chemotherapy, but these methods have greater side effects [13,14].

The treatment of breast cancer with traditional Chinese medicine (TCM) has long been recorded in ancient books of TCM, and TCM treatment of TNBC has its unique advantages [15,16]. Our previous study revealed that *Prunella vulgaris* has obvious anti-breast cancer effects [17,18]; however, there have been no in-depth studies of its anti-TNBC mechanism. Triterpenoid compounds are the main active ingredients of *Prunella vulgaris* that exhibit anti-breast cancer activity [19]. Modern studies have shown that the main chemical components of *Prunella vulgaris* include triterpenoids, phenolic acids and flavonoids [20], which have certain antitumor, blood-pressure- and blood-lipid-lowering, detumescence, immunomodulation and anti-inflammation effects [19,21,22,23,24]. Studies have shown that the extract of *Prunella vulgaris* can effectively inhibit the proliferation and migration of breast cancer cells and promote their apoptosis [25,26]. A series of triterpenoids of *Prunella vulgaris* were isolated, and an activity screening experiment revealed that betulinic acid, 2α-hydroxy-ursolic acid and ursolic acid had obvious inhibitory effects on MCF-7 and MDA-MB-231 breast cancer cells, were nontoxic to the normal breast epithelial cell line MCF-10A and selectively inhibited tumor cell proliferation, migration and the promotion of apoptosis [17,27,28]. However, there are only a few reports on the anti-breast cancer mechanism of *Prunella vulgaris*.

A network pharmacology study revealed that the main triterpenoid components of *Prunella vulgaris*, such as oleanolic acid, ursolic acid and betulinic acid, can target tyrosine phosphatase 1B (PTP1B), and PTP1B is a potential target for breast cancer treatment [29,30]. Overexpression of PTP1B can activate the PI3K-AKT pathway, whereas inhibition of PTP1B can block the PI3K-AKT signaling pathway and inhibit tumor growth [31]. In addition, traditional Chinese medicine is characterized by multiple components, multiple targets and multiple pathways; therefore, we explored another mechanism. In the treatment of TNBC, inhibition of its metastasis is an effective strategy to control its development. Studies have shown that ursolic acid can inhibit the metastasis of prostate cancer and human thyroid papillary carcinoma by inhibiting the CXCL12/CXCR4 signaling axis [32,33]. The CXCL12/CXCR4 signaling axis is regulated by the tumor suppressor gene *Il-24* and is strongly associated with breast cancer [34,35,36,37]. This study revealed that PVT can effectively upregulate *Il-24* through cell transcriptomics studies. The aim of this study was to investigate the anti-TNBC effects and mechanism of action of total triterpenes from *Prunella vulgaris* (PVT). The study design process is detailed in Figure 1.

## 2. Results

### 2.1. Identification of PVT and Determination of Ursolic Acid and Oleanolic Acid Contents

A total of 1000 g of *Prunella vulgaris* was extracted and separated to yield 11 g of total triterpenoid fraction of *Prunella vulgaris* (PVT), with a yield of 1.1%. The triterpene content in PVT was determined by ultraviolet spectrophotometry. Taking the oleanolic acid concentration (C1) as the abscissa and the absorbance (A1) of oleanolic acid as the ordinate, the linear regression equation was A1 = 1.0951C1 − 0.0348. There was a good linear relationship between the oleanolic acid content and the absorbance in the range of 0.05–0.35 mg·mL^−1^ (R^2^ = 0.9969). The total triterpenoid content of PVT was 64.25%.

After mass spectrometry analysis of PVT, 19 compounds were identified by comparing mass spectrum information such as the ratio of the compound charge, the secondary fragment ions and the cracking rules of the compounds in negative ion mode. The chemical composition is shown in Table 1. The following nine compounds were triterpenoids: 18α glycyrrhetinic acid (a), maslinic acid (b), hederagenin (c), cis-p-coumaroylcorosolic acid (d), oleanolic acid acetate (e), betulinic acid (f), oleanolic acid (g), ursolic acid (h), and liquid ambaric acid (i). A total ion flow diagram and the triterpenoid structure of the triterpenes are shown in Figure 2A.

The contents of two representative triterpenes (oleanolic acid and ursolic acid) in PVT were detected by HPLC, and an HPLC diagram of PVT is Shown in Figure 2B. The oleanolic acid content was 0.064 mg·mL^−1^, which accounted for 11.32% of the total contents of PVT. The content of ursolic acid was 0.195 mg·mL^−1^, which accounted for 34.51% of the total contents of PVT.

### 2.2. Network Pharmacology Analysis of Effects of PVT on TNBC

In this study, a total of 152 potential targets for nine triterpenoids of PVT were identified based on the TCMSP and Swiss Target Prediction databases. A “drug–active-ingredient–target” network diagram was constructed by using Cytoscape software 3.10.1 (Figure 3A). Through searches in the GeneCards and OMIM databases, genes that encoded proteins with correlation scores higher than average were screened, and 809 TNBC targets were obtained after combining the duplicate values. Matching triterpenoid targets with disease (TNBC)-related targets revealed 19 common targets in a Venn diagram (Figure 3B). A Protein–Protein Interaction Network (PPI) network diagram of the above 19 targets was constructed using the STRING database (Figure 3C), and 19 nodes and 60 edges were identified.

After topological analysis, the target numbers of the nine triterpenoids in PVT were M1:14, M2:9, M3:8, M4:1, M5:4, M6:6, M7:10, M8:7 and M9:10 (Table 2). Among them, all nine triterpenoids can act on PTP1B, seven triterpenoids can act on PTPN2 and AR, and six triterpenoids can act on PPARA, FABP4 and ESR2. In view of these findings, it was speculated that PTP1B may be a key target of PVT in triple-negative breast cancer (TNBC), which was verified by subsequent relevant studies (Figure 3D). Metascape was used for KEGG analysis of 19 targets, and the enrichment of KEGG pathways revealed that these targets were closely associated with pathways in cancer, thePI3K–Akt signaling pathway, lipids and atherosclerosis (Figure 3E).

### 2.3. PVT Inhibited the Proliferation, Migration and Invasion of MDA-MB-231 Cells

As shown in Figure 4A,B, PVT exhibited efficient cytotoxicity in MDA-MB-231 cells, and the inhibitory effect of PVT on the cells was concentration-dependent. The IC_50_ values of PVT were 89.9 μg·mL^−1^, 42.4 μg·mL^−1^ and 34.1 μg·mL^−1^ for 24, 48 and 72 h, respectively. However, PVT had no obvious toxicity against MCF-10A cells. These results indicated that PVT had the ability to selectively target cancer cells. Furthermore, the colony formation capabilities of MDA-MB-231 cells were distinctly inhibited by treatment with low, medium and high concentrations of PVT (Figure 4C–E). In addition, compared with that in the Ctrl group, the number of cells that entered the lower compartment was significantly reduced in the PVT-10, PVT-20 and PVT-40 groups, and the area occupied by invading cells was also significantly reduced. These findings indicate that PVT can effectively inhibit the invasion of MDA-MB-231 cells (Figure 4F–H). Compared with the control group, the wound-healing ability of MDA-MB-231 cells decreased markedly after treatment with medium and high concentrations of PVT, indicating that PVT can inhibit the migration of MDA-MB-231 cells.

### 2.4. PVT Effectively Promoted the Apoptosis of MDA-MB-231 Cells

To investigate the effect of PVT on the apoptosis of MDA-MB-231 cells, Hoechst33242 fluorescence staining was performed to observe apoptosis. Compared with those in the Ctrl group, the number of apoptotic cells in the PVT group showed obvious nuclear collapse, nuclear fragmentation, irregular nuclear staining and even hyalinoid lesions, indicating that PVT could induce apoptosis in MDA-MB-231 cells (Figure 5A,B).

The effect of PVT on the apoptosis of MDA-MB-231 cells was also studied by flow cytometry. Compared with that of the Ctrl group, the numbers of apoptotic cells in the Taxol and PVT groups were significantly increased, especially the number of late apoptotic cells (upper right/QI-UR), and the number of PVT concentration groups was concentration-dependent (*p* < 0.05, *p* < 0.001). These results indicated that PVT could significantly promote the apoptosis of MDA-MB-231 cells (Figure 5C,D).

Tumors are a type of cell cycle disease. The cell cycle plays an important role in regulating tumor cell growth. After PVT was applied to the MDA-MB-231 cells, the cells gradually changed from fusiform to round, and some of them floated. The cells were then collected for flow cytometry. As shown in Figure 5E,F, the number of MDA-MB-231 cells in the G1 phase significantly increased, whereas the numbers of cells in the S and G2 phases significantly decreased, indicating that MD-MB-231 cells were blocked in the G1–S phase and that PVT affected the normal mitosis of tumor cells.

Bax and Bcl-2 are apoptosis-related factors. The Bax protein is a pro-apoptotic member of the Bcl-2 family, and Bcl-2 is an apoptotic protein that controls cell differentiation, growth and apoptosis. WB was used to detect the effect of PVT on the expression levels of Bax and Bcl-2 in MDA-MB-231 cells. The results (Figure 5G–J) revealed that, compared with that in the Ctrl group, Bax protein expression was upregulated in the PVT groups, whereas Bcl-2 protein expression was significantly decreased, and the difference was statistically significant (*p* < 0.05 or *p* < 0.01). These results indicated that PVT decreased the protein expression of Bcl-2 and increased the protein expression of Bax in MDA-MB-231 cells.

### 2.5. Transcriptomic Analysis of Effects of PVT on MDA-MB-231 Cells

To systematically investigate the biological processes underlying the inhibition of breast cancer cell proliferation by PVT, MDA-MB-231 cells treated with PVT or not were sequenced by transcriptome sequencing. The results of the differential expression analysis revealed that a total of 1673 genes were upregulated and 1685 genes were downregulated in the MDA-MB-231 cells (Figure 6A). KEGG enrichment analysis revealed that the most significant pathways with differential gene expression included the cell cycle, DNA replication and related pathways (Figure 6B). A qRT-PCR assay was used to verify the top five downregulated and upregulated genes (Table 3) in the transcriptomic results, which revealed that PVT significantly upregulated the mRNA expression levels of *Il24* and *Frmpd3* and downregulated the mRNA expression levels of *Serpina5*, *Tnc*, *Cxcl10*, *Neurl1b*, *Lgr5* and *Psg8* (Figure 6C); these results are consistent with those of the transcriptome analysis section.

### 2.6. PVT Effectively Inhibited the Growth of TNBC Tumors

The anti-TNBC effect of PVT was observed in nude mice bearing tumors. The animal experimental design is shown in Figure 7A. According to the tumor growth curve (Figure 7B), the tumor volume reached approximately 80 mm^3^ after 21 days of tumor cell inoculation. After one week of drug intervention, the tumor growth rates of nude mice in all drug administration groups were compared with that in the Mod group, which revealed that the difference in tumor size between the drug administration group and the Mod group further increased with increasing intervention time. Both Taxol and PVT could inhibit breast tumor growth.

After the tumors were established on a body map (Figure 7C), the tumor sites of nude mice were scanned by micro-CT. A two-dimensional transverse section map was established, and a three-dimensional model was constructed (Figure 7D).

The tumors were mainly implanted in the second pair of breast fat pads on the left side. Compared with those of the Mod group, the tumors of the nude mice in each drug administration group were significantly reduced, and calcification had occurred inside the tumors. Both Taxol and PVT could effectively inhibit tumor growth. The tumors of the nude mice were round or oval, with irregular protrusions on some tumors. The results (Figure 7E–G) showed that the size and weight of tumors in the Mod group were greater than those in the other groups (*p* < 0.05, *p* < 0.01), indicating that PVT had an anti-TNBC effect. The pathological changes were observed by HE staining. The results (Figure 7H) showed that the tumor cells in the Mod group were densely arranged and that the tumor cells in the Mod group grew vigorously, with many nuclear divisions observed. Compared with those of the Mod group, the tumor tissues of the Taxol, PVTH and PVTM + Taxol groups presented large necrotic areas, loose tumor tissues, irregular cell arrangement, significantly reduced cell numbers and cytoplasmic vacuolation. The tumor tissues of nude mice in the PVTL and PVTM groups were loose, and the tumor cells were sparsely arranged, with tumor nucleus fragmentation or contraction, partial cell necrosis and inflammatory cell infiltration observed, indicating that Taxol and PVT could inhibit the proliferation of tumor cells and had antitumor effects.

### 2.7. PVT Promoted Tumor Apoptosis in TNBC Model Mice

The apoptosis of tumor cells in situ was detected by TUNEL fluorescence staining. As shown in Figure 8A,B, there was little green fluorescence in the Mod group, indicating that the number of apoptotic cells was very low. Compared with those in the Mod group, there was a large amount of green fluorescence and there were no nuclei in some parts of the tumor tissue in the Taxol group. The numbers of TUNEL-positive cells in the PVTL, PVTM, PVTH and PVTM + Taxol groups were significantly greater, and the apoptosis indices were significantly greater, indicating that both PVT and Taxol could promote the apoptosis of tumor cells. In addition, detection of the Bax and Bcl-2 proteins in tumor tissues by WB (Figure 8G–J) and immunofluorescence (Figure 8C–F) revealed that PVT effectively upregulated the proapoptotic protein Bax and downregulated the antiapoptotic protein Bcl-2.

The protein expression levels of Bax and Bcl-2 in the tumor tissues of nude mice were measured by Western blot. Representative Western blot bands and the quantification of protein expression levels are shown (*n* = 3).

### 2.8. Effects of PVT on PTP1B and PI3K/AKT/mTOR Signaling Pathways in TNBC Cells and Tumor Tissues

The qRT-PCR results (Figure 9A) revealed that PVT effectively decreased the mRNA expression levels of *ptp1b*, *pi3k*, *akt* and *mtor* in MDA-MB-231 cells, and the expression levels of the relative proteins were further detected by WB and immunofluorescence. PVT effectively downregulated the protein expression levels of PTP1B, PI3K, AKT and mTOR in MDA-MB-231 cells (Figure 9B–J). Similarly, the detection of corresponding proteins in the tumor tissues of nude mice with TNBC revealed that PVT effectively downregulated the protein expression levels of PTP1B, PI3K, AKT and mTOR in tumor tissues (Figure 9K–O).

### 2.9. Silencing PTP1B Can Play an Anti-TNBC Role

siRNA was used to silence PTP1B, and siR2 was selected as a follow-up research tool after screening (Figure 10A–D). The results of the present study (Figure 10E–K) revealed that siR significantly reduced the protein expression of PTP1B and that both PVT and siR combined with PVT significantly reduced the protein expression of PTP1B. After silencing PTP1B or downregulating PTP1B, the protein levels of PI3K, AKT and mTOR also decreased, whereas PVT and PVT combined with siR had stronger downregulatory effects on these three proteins. In addition, silencing PTP1B could upregulate Bax and downregulate Bcl-2. PVT and PVT combined with siR had more obvious regulatory effects on Bax and Bcl-2. These results suggest that PVT can inhibit the PI3K/AKT/mTOR signaling pathway against TNBC by blocking PTP1B.

### 2.10. PVT Inhibited the Metastasis of TNBC by Regulating the IL-24/CXCL12/CXCR4 Signaling Axis

WB was used to detect the protein expression levels of IL-24, CXCL12 and CXCR4 in MDA-MB-231 cells treated with or without PVT. The results (Figure 11A–D) showed that, compared with that in the Ctrl group, the protein expression levels of IL-24 in the Taxol and PVT groups at different concentrations were significantly greater. Compared with that in the Ctrl group, the CXCL12 protein levels in the Taxol and PVT groups were significantly lower. Compared with that in the Ctrl group, the CXCR4 protein levels in the PVT-20 and PVT-40 groups were significantly decreased.

The results (Figure 11E–H) revealed that, compared with the Mod group, the IL-24 protein levels in the tumor tissues of the PVTM, PVTH, Taxol and PVTM + Taxol groups were increased, whereas the CXCL12 protein level was decreased. Compared with those in the Mod group, the CXCR4 protein levels in the tumor tissues of the PVT, Taxol and PVTM + Taxol groups were downregulated.

## 3. Discussion

TNBC is a type of breast cancer characterized by high heterogeneity. Owing to the lack of effective therapeutic targets, surgical resection, radiotherapy and chemotherapy are still used in clinical treatment. However, TNBC still has high recurrence and mortality rates. The toxic side effects of chemoradiotherapy are great, which seriously affects the quality of life of patients. Thus, the development of new therapeutic strategies to combat TNBC is urgently needed.

Because of its unique antitumor activity and few side effects, Chinese medicine has become a hot topic in the field of tumor research in recent years [15,38].

In this study, the effects of triterpene of *Prunella vulgaris* (PVT) on TNBC were verified through in vivo and in vitro studies, and the mechanism by which PVT affects TNBC was explored through network pharmacology and transcriptomics analysis and then verified by molecular biology techniques.

Promoting the apoptosis of tumor cells is one of the main ways for drugs to exert antitumor effects. In this study, it was found that PVT can effectively promote the apoptosis of triple-negative breast cancer cells in vitro and in vivo, and the upregulation effect of PVT on Bax even exceeded that of Taxol; the possible reason is that PVT contains various compounds. Studies have shown that ursolic acid, oleanolic acid, betulinic acid and liquid ambaric acid can induce apoptosis of breast cancer cells by regulating Bax and Bcl-2 [39,40,41,42]. Maslinic acid and hederagenin can induce the apoptosis of other tumor cells by downregulating the expression of Bcl-2 and upregulating that of Bax [43,44,45,46]. Therefore, the synergistic effect of multiple components on the regulation of Bax is stronger. The effect of Taxol on inducing the apoptosis of MDA-MB-231 cells was stronger than that of PVT, but the regulatory effect on Bax was weaker than that of PVT at a high dose, which may be because Taxol also induces apoptosis involving the activation and regulation of various molecular signaling pathways, which includes an imbalance in Bcl-2 family proteins, the activation of the Caspase cascade reaction, a reduction in mitochondrial membrane potential, the fragmentation of DNA, etc. [47,48].

Through network pharmacology studies, the targets of nine triterpenes in PVT were found to be closely related to cancer pathways and the PI3K-Akt signaling pathway, and all nine triterpenes could target PTP1B. PTP1B is a negative regulator of insulin signal transduction. By dephosphorylating tyrosine residues on the insulin receptor, the insulin receptor cannot bind to insulin, thus blocking the activation of the downstream PI3K/Akt signaling pathway and triggering insulin resistance [49]. PTP1B is also a potential target for breast cancer treatment [30,31] and is overexpressed in TNBC and ER+ breast cancer [50]. PTP1B can be used as an immune checkpoint to participate in the immune function of T cells in vivo, and the deletion of PTP1B can significantly increase the numbers of various immune cells, especially T cells [51]. In addition, PTP1B regulates the PI3K/AKT/mTOR signaling pathway [52]. The PI3K/AKT/mTOR pathway is one of the most frequently dysregulated signaling pathways in cancer and plays central roles in the growth, proliferation, motility and survival of tumor cells. Studies have shown that the PI3K/Akt/mTOR signaling pathway is involved in metabolic processes such as glucose uptake and glycolysis, which are closely related to the occurrence and development of breast cancer, indicating that the PI3K/AKT/mTOR pathway should be targeted for the treatment of TNBC [53].

This study revealed that nine triterpenoids in PVT can target PTP1B and that the target pathway of triterpenoids is related to the PI3K–Akt signaling pathway, suggesting that PVT can play an anti-TNBC role through the PTP1B/PI3K/AKT/mTOR signaling pathway.

Generally speaking, transcriptomic analysis is used for large-scale screening to reflect the overall trend of gene expression changes in samples, but it cannot guarantee that the trend of every gene’s expression change is consistent with that of qPCR. Therefore, the results of transcriptomics should be validated in a targeted way [54]. Transcriptomic studies have shown that PVT can increase the level of *Il-24* mRNA and downregulate the level of *Cxcl10* mRNA in MDA-MB-231 cells, whereas IL-24 can inhibit the growth of various tumor cells, including breast cancer cells, and induce the apoptosis of tumor cells [55,56]. In addition, IL-24 can exert antitumor effects through CXCR4/CXCL12, and CXCL10, CXCL12 and CXCR4 are highly expressed in breast cancer tissues [57]. The CXCL12/CXCR4 axis mainly plays biological roles through the PI3K/AKT/mTOR signaling pathway [58]. These results suggest that PVT may inhibit the CXC12/CXCR4 signaling axis through the upregulation of IL-24 and then inhibit the PI3K/AKT/mTOR signaling pathway, thus playing an antitumor role.

The PI3K/AKT signaling pathway is one of the most frequently activated pathways in human tumors and can promote tumor proliferation and migration through the regulation of tumor metabolism. Through molecular biological verification, it was found that PVT can upregulate the IL-24 and Bax proteins and downregulate PTP1B, PI3K, AKT, mTOR, CXCL12, CXCR4, Bcl-2 and other proteins. It can inhibit the proliferation of tumor cells, promote their apoptosis and inhibit the metastasis of tumor cells to achieve antitumor effects.

Thus, the mechanism of the anti-TNBC effect of PVT may include, but is not limited to, the regulation of the PTP1B/PI3K/AKT/mTOR signaling pathway and the IL-24/CXCL12/CXCR4 signaling axis, which reflects the characteristics of the multiple-component and multitarget actions of traditional Chinese medicine (Figure 12).

## 4. Materials and Methods

### 4.1. Preparation of PVT

*Prunella vulgaris* (PV) was crushed and sifted from 60 to 80 mesh, and 70% ethanol was added at a 1:10 ratio for reflux extraction for 1 h. This process was repeated three times, and then, the extraction solutions were combined and filtered, and the alcohol was recovered and concentrated to a concentration of 15%.

The sample was transferred to an AB-8 macroporous resin chromatography column and eluted with five times the column volume of pure water, 50% ethanol, 70% ethanol and then pure ethanol, and the pure ethanol elution mixture was collected, concentrated and freeze-dried with a freeze-dryer (Riken Machinery Co., Ltd., Tokyo, Japan) to obtain the PVT freeze-dried powder for use.

The total triterpene content was determined by ultraviolet spectrophotometry. First, 10.29 mg of oleanolic acid standard was accurately weighed and diluted in 10 mL of anhydrous methanol to the scale line, shaken to obtain a standard stock solution with a concentration of 1.029 mg·mL^−1^, and then diluted with anhydrous methanol to obtain standard solutions with concentrations of 0.206, 0.257, 0.309, 0.36 and 0.412 mg·mL^−1^. Then, 0.2 mL of oleanolic acid solution of various concentrations was accurately removed and added to 10 mL test tubes with plugs, and 0.2 mL of 5% vanillaldehyde–acetic acid was added to each test tube and shaken for 10 s, 1.8 mL perchloric acid was added, and the mixture was shaken for 10 s and then placed in a water bath at 65 °C for 25 min. Then, the mixture was quickly cooled in ice water, 2.5 mL glacial acetic acid was added, and the 200 μL solution was shaken for 10 s. Then, the 200 μL solution was added to a 96-well plate, and the absorption value (A) was detected at 547 nm by a microplate reader (Synergy HTX, BioTek, Winooski, VT, USA). Anhydrous methanol was used as a blank control to establish a standard curve. Then, 5.65 mg of the freeze-dried powder of PVT added to methanol was dissolved, the volume was fixed, and a PVT solution with a concentration of 0.565 mg·mL^−1^ was obtained. The absorption value of the components was measured at 547 nm, and the triterpene content in PVT was calculated according to the standard curve.

### 4.2. Chemical Profile of PVT

PVT was characterized by liquid chromatography–mass spectrometry (LC-MS, Agilent 6545 Q-TOF, Agilent, Santa Clara, CA, USA). The contents of ursolic acid and oleanolic acid were determined by high-performance liquid chromatography (HPLC, Waters e2695, Waters, Milford, MA, USA).

Approximately 10 mg of the freeze-dried powder of PVT was dissolved in methanol to achieve a mass concentration of 2 mg·mL^−1^. The solution was centrifuged at 13,000 rpm for 10 min, and the supernatant was obtained for mass spectrometry. The chromatographic conditions were as follows: Agilent Zorbax eclipse plus C18 column (2.1 mm × 50 mm, 1.8 μm); mobile phase: 5 mM ammonium formic acid solution (A)–acetonitrile (B); gradient elution: 15% B (0–5 min), 15–35% B (5–10 min), 35–80% B (10–20 min), 80% B (20–25 min), 80–100% B (25–28 min), 100% B (28–32 min), 100–5% B (32–34 min); a flow rate of 0.3 mL·min^−1^; a wavelength of 330 nm; a column temperature of 30 °C; and an injection volume of 2 μL. The mass spectrum conditions were as follows: negative ion mode, electrospray ionization (ESI), dry gas nitrogen flow rate = 8 L·min^−1^, dry gas temperature = 320 °C, nebulizer pressure = 35 psi, capillary voltage = 3000 V, nozzle voltage = 1000 V, and collision-induced dissociation voltage = 150 V. The skimmer voltage was 65 V, and the data acquisition (*m*/*z*) range was 50–1100. MassHunter Acquisition Software 13.0 and MassHunter Workstation Software V b.04.00 (Agilent, Santa Clara, CA, USA) were used for data acquisition and processing, respectively.

A total of 10.29 mg of oleanolic acid and 9.91 mg of ursolic acid were accurately weighed and dissolved in methyl alcohol to prepare standard stock solutions. Aliquots of 50, 100, 150, 200 and 350 μL of the standard stock solutions were moved into 1 mL volumetric flasks and mixed. The concentrations of the standard solutions were approximately 0.05, 0.10, 0.15, 0.20, 0.25 and 0.35 mg·mL^−1^. A total of 5.65 mg of the freeze-dried powder of PVT was accurately weighed and dissolved in methyl alcohol to prepare a sample solution of PVT with a mass concentration of 0.565 mg·mL^−1^. The chromatographic conditions were as follows: Agilent ZORBAX SB-C18 column (4.6 × 250 mm, 5 μm); mobile phase: methanol (A) and 0.1% formic acid solution (B) (equal elution: 85% A, 15% B); flow rate: 1.0 mL·min^−1^; wavelength: 210 nm; column temperature: 30 °C; injection volume: 10 μL.

### 4.3. Network Pharmacology Analysis of PVT in the Treatment of TNBC

The effective components and action targets of PVT were retrieved from the TCMSP (https://old.tcmsp-e.com/tcmsp.php, accessed on 3 April 2024), DrugBank, and STITCH (http://stitch.embl.de/, accessed on 3 April 2024) databases; 9 effective components and 152 action targets were obtained. From GENECARD (https://www.genecards.org/, accessed on 3 April 2024) and OMIM (https://www.omim.org, accessed on 3 April 2024), 809 TNBC targets were retrieved. Venny2.1.0 (https://bioinfogp.cnb.csic.es/, accessed on 3 April 2024) was used to analyze the intersection of the active ingredient targets of PVT and TNBC, and 19 common targets were obtained. The common targets were imported into the STRING database (confidence level > 0.9) (https://cn.string-db.org/, accessed on 3 April 2024), and PPI analysis was performed. Cytoscape 3.7.2 software (https://cytoscape.org/, accessed on 3 April 2024) and the NetworkAnalyz plugins were used to construct and visualize the network of “PVT–active ingredients–potential TNBC targets”, and the common targets were imported into the Metascape database (https://metascape.org/gp/index.html#/main/step1, accessed on 3 April 2024), and KEGG pathway enrichment analyses were performed (Species: Homo sapiens, *p* < 0.05).

### 4.4. Cell Viability Assay

The cell viabilities of MDA-MB-231 cells and MCF-10A cells were tested using a Cell Counting Kit-8 (CCK8) assay. The cells were seeded into 96-well plates (1 × 10^4^/well, 100 μL/well) and incubated with different concentrations of PVT (0–160 μg·mL^−1^) or Taxol (50 nM) for 24, 48 or 72 h. CCK8 (10 μL) was added, and the mixtures were incubated for 4 h. Cell viability was measured based on the reduction of CCK8 dye in living cells to yellow formazan crystals at an optical density of 450 nm, and the survival rate of the cells was calculated. Cells cultured in the medium were used as negative controls.

### 4.5. Cell Proliferation Assay

A colony formation assay was performed to investigate the effects of PVT on cell proliferation ability. MDA-MB-231 cells were seeded into a 6-well plate (4 × 10^3^ cells/well, 2 mL/well) and cultured overnight. Then, the cells were treated with different concentrations of PVT (10, 20, 40 μg·mL^−1^) or Taxol (50 nM) for 14 days, and the culture medium containing PVT or Taxol was changed every 3 days. One milliliter of 4% paraformaldehyde was added to each well for 30–60 min.

Then, 1 mL of crystal violet dye was added to each well for 15 min, after which images were taken, and the number of colonies was counted. The proliferative potential was determined by the number of colonies generated from a single cell. Colonies with diameters larger than 2 nm were counted.

### 4.6. Scratch Assay

A scratch assay was used to explore the effect of PVT on the migration ability of breast cancer cells. MDA-MB-231 cells were prepared and transplanted into a 24-well plate (4 × 10^4^ cells/well, 400 μL/well). Once confluence was achieved, a 10 μL pipette tip was used to perform the scratching operation. Subsequently, the plates were carefully rinsed with phosphate-buffered saline (PBS) before adding complete medium with different concentrations of PVT (10, 20, 40 μg·mL^−1^) or Taxol (50 nM). Scratch recovery was documented after 0, 12, 24, 48 and 72 h of scratching using an inverted microscope. Finally, the degree of scratch closure was calculated based on the wound-healing area via ImageJ_v1.8.0 software with the following calculation formula: cell migration rate = (initial scratch area − final scratch area)/initial scratch area × 100%.

### 4.7. Transwell Migration Assay

Transwell migration assays were used to explore the effects of PVT on the invasion ability of breast cancer cells. Approximately 1 × 10^5^ MDA-MB-231 cells were cultured in Matrigel-coated upper chambers containing specified concentrations of PVT (10, 20, 40 μg·mL^−1^) or Taxol (50 nM), and 600 μL RPMI 1640 cell medium was injected into the lower chamber. After 24 h, the cells were stained with 0.1% crystal violet at room temperature for 30 min, washed twice with PBS and observed under a microscope. The number of cells in 10 fields was counted, and images were taken.

### 4.8. Flow Cytometry

Apoptosis and the cell cycle were detected by flow cytometry. MDA-MB-231 cells were seeded into a 6-well plate (1 × 10^5^ cells/dish, 2 mL/dish). The cells were treated with different concentrations (10, 20, 40 μg·mL^−1^) of PVT or Taxol (50 nM) for 48 h, collected and washed three times with ice-cold PBS. The cells were then resuspended in binding buffer and then stained with Annexin V-FITC and propidium iodide (PI) in accordance with the manufacturer’s guidelines. Finally, apoptotic cells were detected by flow cytometry (A00-1-1102, Beckman, Brea, CA, USA).

### 4.9. Cell Transcriptomics

#### 4.9.1. RNA Sequencing

MDA-MB-231 cells were treated with a vehicle or 40 μg·mL^−1^ PVT for 48 h. The cells were collected and washed with cold PBS three times. Total RNA was extracted according to the instructions provided with the RNA purification kit, and the RNA was resuspended in diethyl pyrocarbonate. After its integrity and purity were determined, the RNA was subjected to PCR amplification (Rotor-Gene RG-3000, QIAGEN, Hilden, Germany) for the construction of a cDNA library. Raw reads were filtered to exclude low-quality sequences that might affect the data quality and subsequent analysis.

The cDNA library was sequenced by high-throughput sequencing (HiSeq™ 2000, Illumina, San Diego, CA, USA). The raw sequence data were filtered using SOAPnuke (v1.5.6) to remove low-quality sequences and adaptor reads. Then, HISAT2 (v2.1.0) software was used to map the clean reads to the reference genome using standard mapping parameters. RSEM (v1.3.1) software was used to quantify gene expression. DESeq2 (v1.4.5) was used for in-group differential gene detection under the conditions of a fold change ≥ 2 and an adjusted *p* value ≤ 0.001. Differences between groups were analyzed with PoissonDis under the conditions of a fold change ≥ 2 and FDR ≤ 0.001. Differential genes were verified by qRT-PCR (Roche, LightCycler 96).

#### 4.9.2. Quantitative Real-Time PCR

A qRT-PCR assay was used to investigate the effects of PVT on related mRNA expression in breast cancer cells. MDA-MB-231 cells were inoculated in cell-culture dishes (1 × 10^6^ cells/dish, 5 mL/dish). After treatment with the different concentrations of PVT (10, 20, 40 μg·mL^−1^) or Taxol (50 nM) for 48 h, the cells were collected. Total mRNA was extracted using TRIzol Reagent according to the manufacturer’s protocol, and the mRNA was then reverse-transcribed into cDNA. The primer sequences for *Lgr5*, *Agt*, *Psg8*, *Il24*, *Kcp*, *Serpina5*, *Tnc*, *Cxcl10*, *Frmpd3*, *Neurl1b*, *Ptp1b*, *Pi3k*, *Akt*, *mtor* and *Gapdh* are listed in Table 4. The gene expression levels of the target genes were normalized to that of *Gapdh* and calculated using the comparative CT value method (2^−△△CT^).

### 4.10. Hoechst 33342 Staining

Hoechst 33342 fluorescence staining was used to visualize the nuclear morphological changes in MDA-MB-231 cells exposed to PVT. MDA-MB-231 cells were seeded into 6-well plates with a climbing piece at a density of 1 × 10^5^ cells/well and treated with different concentrations of PVT (10, 20, 40 μg·mL^−1^) or Taxol (50 nM) for 24 h, after which the cells were fixed with 4% paraformaldehyde solution for 10 min, washed twice with PBS, stained with Hoechst 33342 dyeing solution for 5 min at room temperature in the dark, and then observed and photographed by fluorescence microscopy.

### 4.11. Triple-Negative Breast Cancer Mouse Model Construction and Treatment

After one week of adaptive feeding, mice were used to construct the TNBC model. MDA-MB-231 cells were presuspended in sterile PBS, and then, 100 μL of the cell suspension (1 × 10^8^ cells/mL) was injected into the second pair of breast fat mammary glands on the left side of each mouse [59,60]. When the tumor volume had reached approximately 80 mm^3^, 36 tumor-bearing nude mice were randomized into six groups—Mod group (0.2 mL/pure water, *i.g.* (gavage), qd (once a day)), Taxol group (Taxol, 20 mg·kg^−1^, *i.p.* (intraperitoneal injection), every 5 days), PVTL group (PVT, 50 mg·kg^−1^, *i.g.*, qd), PVTM group (PVT, 100 mg·mg·kg^−1^, *i.g.*, qd); PVTH group (PVT, 50 mg·kg^−1^, *i.g.*, qd); PVTM + Taxol Group (PVT, 100 mg·kg^−1^, *i.g.*, qd; Taxol, 20 mg·kg^−^1, every 5 days)—and the mice were treated for 21 days. The tumor volume (V = 0.5 × LW^2^, where L and W are the tumor length and width, respectively) and body weights were measured and recorded every 4 days. At the end of the experiment, the eyeballs of the mice were removed for blood collection. Tumors from each group were isolated, photographed, weighed and divided into 2 parts, one of which was stored at low temperature for protein detection, and the other fixed in 4% paraformaldehyde and used for pathological detection.

### 4.12. Micro-CT Scanning

After 21 days of administration, the nude mice were fasted with free access to water overnight. Six nude mice were randomly selected from each group for micro-CT scanning. The animals were anesthetized with isoflurane aerosol before scanning, the limbs were extended and fixed vertically on the stent, and the anesthesia was maintained at a low dose. The micro-CT scanning parameters were as follows: the X-ray tube voltage was 90 V, the tube current was 88 μA; the scanning field of view was 36 mm × 36 mm; the pixel size was 72 μm; the two-phase breathing gating mode was open; the scan area was the breast cancer tumor; the detector mode was high-resolution; and the scanning mode was 4 min.

### 4.13. Hematoxylin and Eosin (H&E) Staining

Pathological changes in the tumor tissues were observed by H&E staining. The fixed tumor tissues were embedded in paraffin and sectioned to 5 μm in thickness. The sections were dewaxed and dehydrated with xylene and gradient ethanol. The nuclei were stained with hematoxylin for 10 min and differentiated with 1% ethanol hydrochloride differentiated for a few seconds, after which 0.6% ammonia was added to return the color to blue. Eosin was used to stain the cytoplasm for 2 min, and the sections were dehydrated with an ethanol gradient and sealed with a neutral dendrimer. The appropriate field of view was selected for photographing.

### 4.14. Western Blotting Analysis

After treatment with specified concentrations of PVT and Taxol, the cells were lysed with precooled RIPA buffer containing a protease inhibitor cocktail and freshly prepared PMSF on ice for 30 min, after which the cells were scraped off with a cell scraper, the samples were collected into centrifuge tubes and centrifuged at 12,000 rpm for 5 min, and the supernatants were collected. The protein concentrations were determined following the instructions provided for the BSA protein assay kit.

The samples were separated by SDS-PAGE on 4–12% gel and transferred to a PVDF membrane. The PVDF membrane was blocked with 5% skim milk at room temperature for 2 h and then incubated overnight at 4 °C with the primary antibody (diluted 1:1000). After washing with TBST, the membrane was incubated for 2 h in a constant-temperature shaker at 37 °C with the HRP-conjugated secondary antibody (diluted 1:5000). After washing with TBST, the membranes were visualized using an enhanced chemiluminescence (ECL) detection kit, and images were captured using a gel imaging system (Bio-Rad ChemiDoc XRS+, Bio-Rad, Hercules, CA, USA).

The tumor tissue samples were cut, and precooled RIPA buffer containing the protease inhibitor cocktail and fresh PMSF were added to the tissue samples, which were then homogenized at 4 °C with a homogenizer until there was no obvious visible solid. The samples were then extracted by ultrasonication in an ice water bath for 10 min and left on ice for 50 min for complete lysis. The next steps were similar to those used for cell protein blotting.

### 4.15. Immunofluorescence Assay

MDA-MB-231 cells were inoculated into 24-well plates with climbing tablets, treated with different concentrations of PVT (10, 20, 40 μg·mL^−1^) or paclitaxel (Taxol, 20 nM) at 24 h, and fixed with 4% paraformaldehyde solution at room temperature. Triton X-100 (0.2%) was added to the cells for 10 min, the cells were blocked with 5% bovine serum albumin (BSA), and monoclonal antibodies (PI3K, AKT, and mTOR) were added. After incubation on a shaking bed at 4 °C overnight, fluorescent secondary antibody was added, and the samples were incubated at 37 °C for 60 min in the dark. Finally, a drop of fluorescent sealer (including DAPI) was added to each dish, and the fluorescence of the target protein was observed and photographed by laser confocal microscopy.

Five-micron-thick tissue sections were prepared, and antigen retrieval was performed using citric acid antigen retrieval buffer (pH 6.0). The primary antibody was then added and incubated overnight at 4 °C. After washing with PBS, the secondary antibody (SA00013-4, 1:500; Proteintech Group, Inc., Wuhan, China) was added and incubated at 37 °C for 60 min, DAPI was used to counter-stain the nuclei, and images were captured under a microscope.

### 4.16. Silencing the PTP1B Gene Using siRNA Technology

MDA-MB-231 cells were divided into a control group (Ctrl), an siRNA-silencing negative control group (Scam), an siRNA-silenced group (siRNA), a drug group (PVT, 40 μg·mL^−1^), and an siRNA-silencing + drug combination group. The siRNA and RNATransmate complexes obtained in the previous experiment were added to the medium, and the MDA-MB-231 cells were cultured in serum-containing medium and incubated for 48 h.

### 4.17. Statistical Processing

SPSS 26.0 software was used to collate the data, and the experimental data are presented as means ± SDs. A *t* test was used for comparisons of the means of two groups, and one-way ANOVA was used for comparisons of the means of multiple groups. *p* < 0.001 indicated that the difference was statistically significant. The experimental data were visualized using GraphPad Prism 8.0 software.

## 5. Conclusions

In summary, the results of our study demonstrate that PVT can effectively inhibit the proliferation, migration and invasion of TNBC cells, promote apoptosis and inhibit tumor growth. The mechanism of action of PVT against TNBC includes, but is not limited to, the inhibition of the PTP1B/PI3K/AKT/mTOR signaling pathway and its regulation of the IL-24/CXCL12/CXCR4 signaling axis.

## Figures and Tables

**Figure 1 ijms-26-01959-f001:**
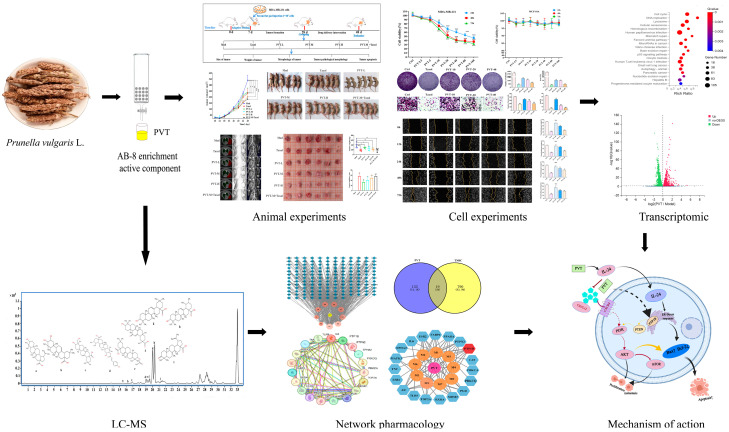
Flowchart of the pharmacology approach for uncovering the effect and mechanism of triterpenes of *Prunella vulgaris* (PVT) in treating TNBC. (* *p* < 0.05, ** *p* < 0.01, *** *p* < 0.001).

**Figure 2 ijms-26-01959-f002:**
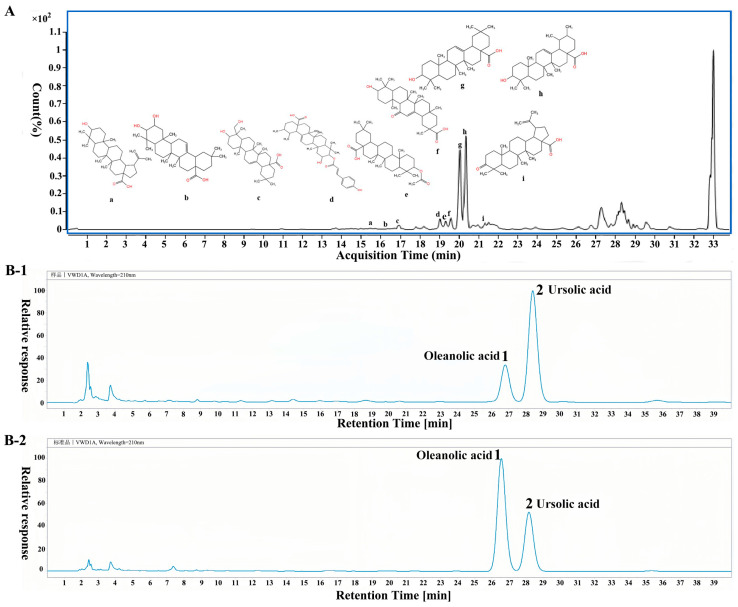
Total ion flow diagrams of PVT and HPLC chromatograms. (**A**) LC-MS profile and chemical compound structures of triterpenoid in PVT; (**B-1**) HPLC spectra of PVT; (**B-2**) HPLC spectra of oleanolic acid and ursolic acid standards.

**Figure 3 ijms-26-01959-f003:**
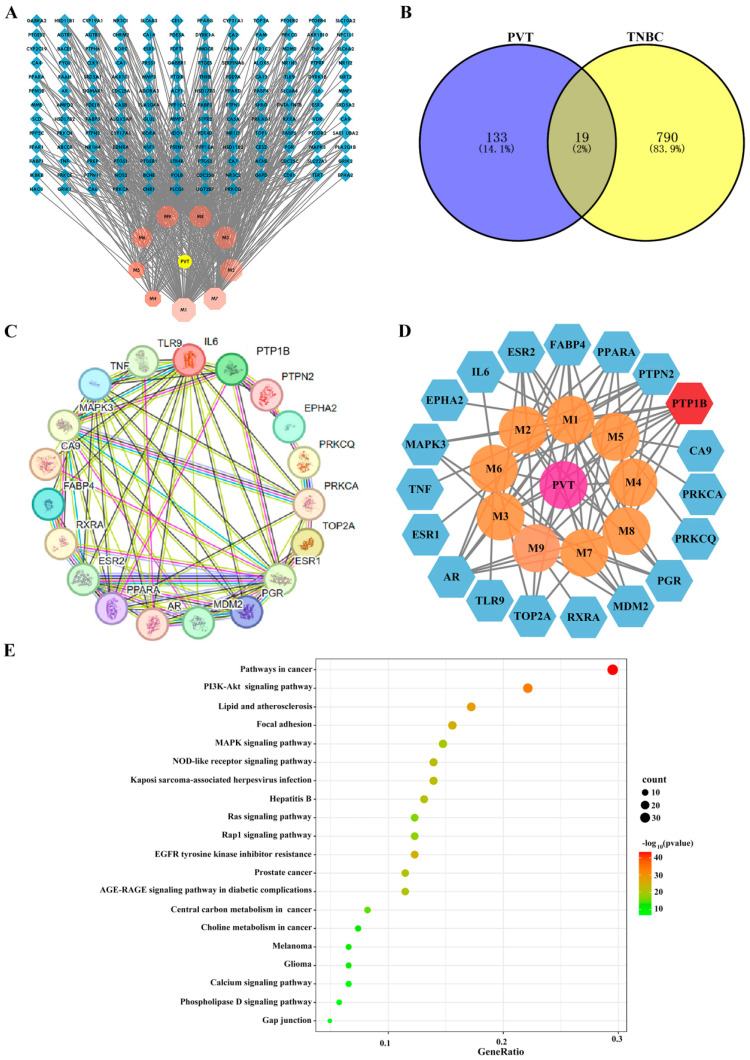
Network pharmacological analysis of PVT against TNBC. (**A**) Target network diagram of 9 triterpenes of PVT; (**B**) Venn diagram of common targets of triterpenes and TNBC; (**C**) PPI network diagram of intersecting targets of triterpenes and TNBC; (**D**) triterpenes and intersection targets network diagram; (**E**) intersection target KEGG pathway enrichment diagram.

**Figure 4 ijms-26-01959-f004:**
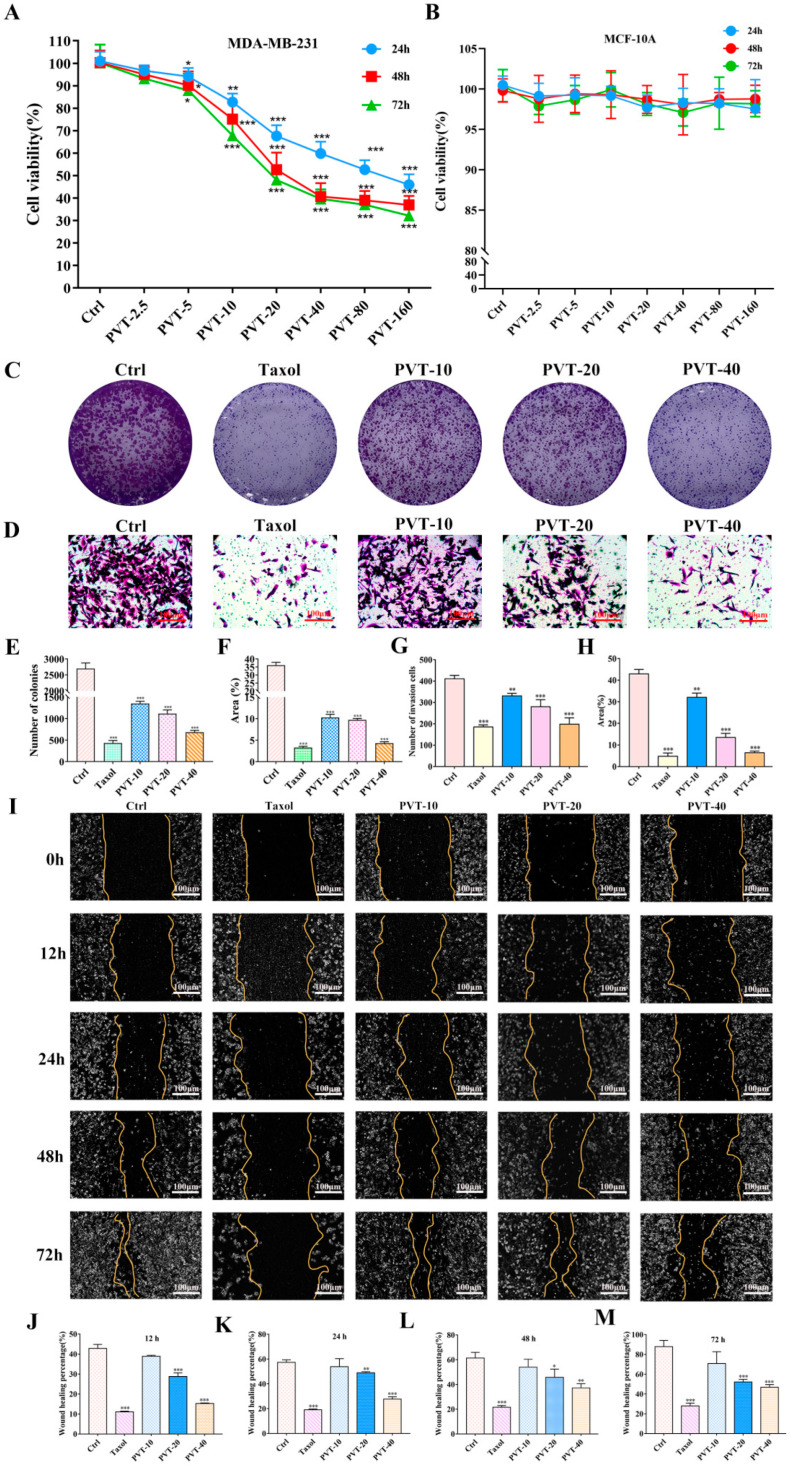
Effects of total triterpenoids (PVT) on proliferation, migration and invasion of MDA-MB-231 cells. (**A**,**B**) Cell viabilities of MDA-MB-231 and MCF-10A were assessed after treated with PVT (0–160 μg·mL^−1^) for 24 h, 48 h and 72 h (mean ± SD, *n* = 5; * *p* < 0.05, ** *p* < 0.01, *** *p* < 0.001); (**C**) effect of PVT on colony formation of MDA-MB-231 cells; (**D**) cell clonal formation statistics; (**E**) percentage of clonal formation area; (**F**) effect of PVT on invasion of MDA-MB-231 cells; (**G**) statistics of number of invasive cells; (**H**) percentage of invasion area; (**I**) effect of PVT on migration of MDA-MB-231 cells; (**J**–**M**) statistical chart of healing area 12 h, 24 h, 48 h and 72 h after administration. Concentration of Taxol in cell experiments was 50 nM. (**E**–**H**,**J**–**M**, mean ± SD, *n* = 3; * *p* < 0.05, ** *p* < 0.01, *** *p* < 0.001).

**Figure 5 ijms-26-01959-f005:**
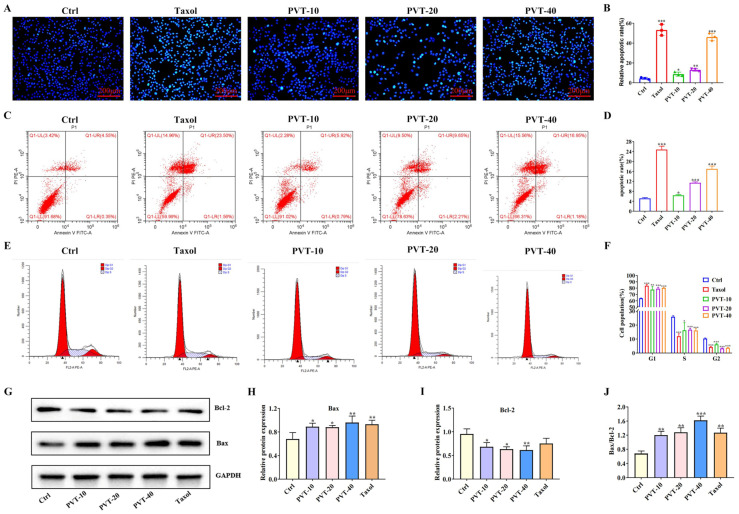
Effect of PVT on apoptosis and cell cycle of MDA-MB-231 cells (mean ± SD, *n* = 3; * *p* < 0.05, ** *p* < 0.01, *** *p* < 0.001). (**A**) Hochest33342 staining; (**B**) apoptosis statistical map; (**C**) apoptosis flow chart; (**D**) apoptosis flow chart; (**E**) cell cycle flow chart; (**F**) cell cycle flow chart; (**G**) apoptosis-related protein Bax and Bcl-2 protein strip chart; (**H**) protein Bax/GAPDH statistical map; (**I**) protein Bcl-2/GAPDH statistical map; (**J**) statistical diagram of protein Bax/Bcl-2. Concentration of Taxol in cell experiments was 50 nM.

**Figure 6 ijms-26-01959-f006:**
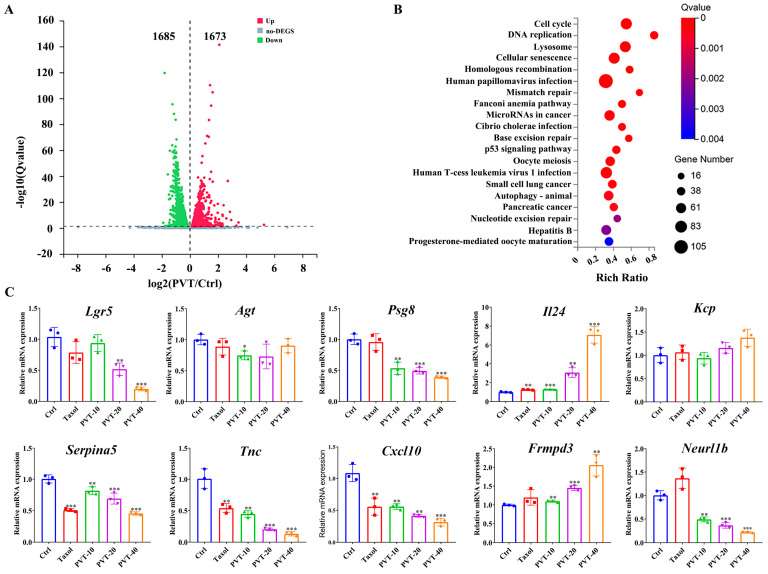
Transcriptomic study of MDA-MB-231 cells and identification of differential genes by qRT-PCR (mean ± SD, *n* = 3; * *p* < 0.05, ** *p* < 0.01, *** *p* < 0.001). (**A**) Volcano map of differential genes; (**B**) KEGG enrichment pathway diagram; (**C**) top five downregulated genes and upregulated genes were detected by qRT-PCR.

**Figure 7 ijms-26-01959-f007:**
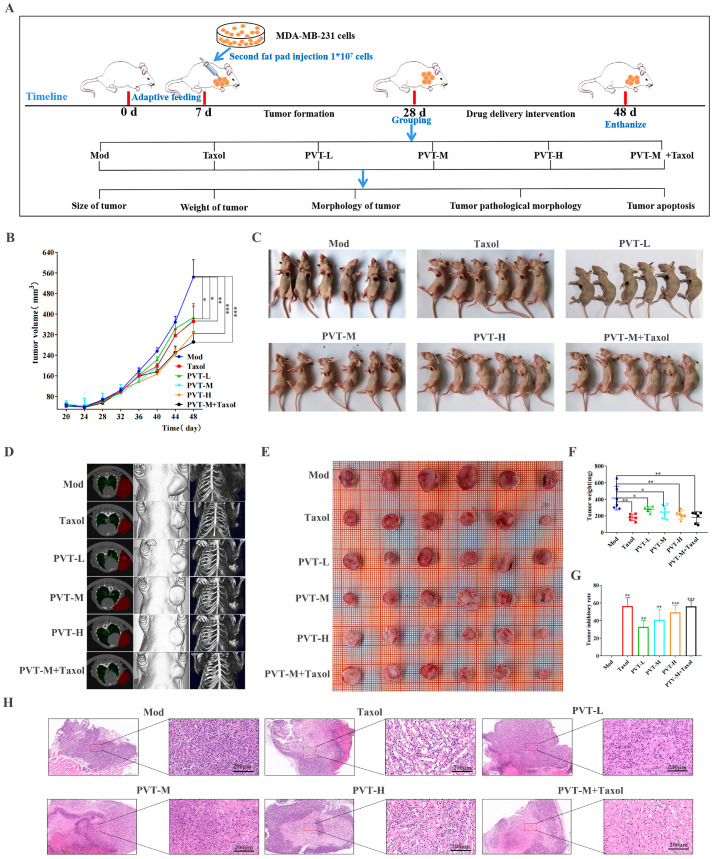
PVT inhibited tumor progression in mice with TNBC (mean ± SD, *n* = 6; * *p* < 0.05, ** *p* < 0.01, *** *p* < 0.001). (**A**) A schematic of the experiment; (**B**) tumor growth curves among different groups; (**C**) tumor-bearing nude mice; (**D**) two-dimensional and three-dimensional images of nude mice were made through micro-CT scanning; (**E**) tumor sizes among different groups; (**F**) tumor weight among different groups; (**G**) statistical graph of drug inhibition rate of tumors; (**H**) representative H&E-stained sections of tumors from different groups. Dosage of Taxol in animal experiments was 20 mg·kg^−1^.

**Figure 8 ijms-26-01959-f008:**
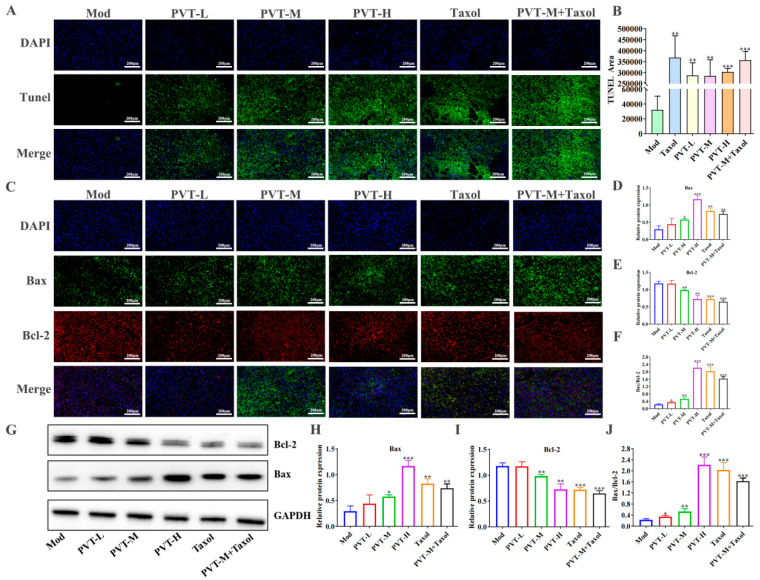
Effect of PVT on apoptosis in tumor tissues (mean ± SD, *n* = 3; * *p* < 0.05, ** *p* < 0.01, *** *p* < 0.001). (**A**) TUNEL staining assay was used to detect cells apoptosis in tumor tissue; (**B**) statistics of TUNEL staining area; (**C**) immunofluorescence staining was used to detect expression of Bax and Bcl-2 in tumor tissues, and images were visualized by confocal microscopy to observe Bax (green), Bcl-2 (red) and nuclei (blue); (**D**–**F**) quantification of immunofluorescence staining for Bax or Bcl-2 or Bax/Bcl-2 in tumor tissues; (**G**) Protein bands of apoptosis-related protein Bax and Bcl-2; (**H**) protein Bax/GAPDH statistical map; (**I**) protein Bcl-2/GAPDH statistical map; (**J**) statistical diagram of protein Bax/Bcl-2.

**Figure 9 ijms-26-01959-f009:**
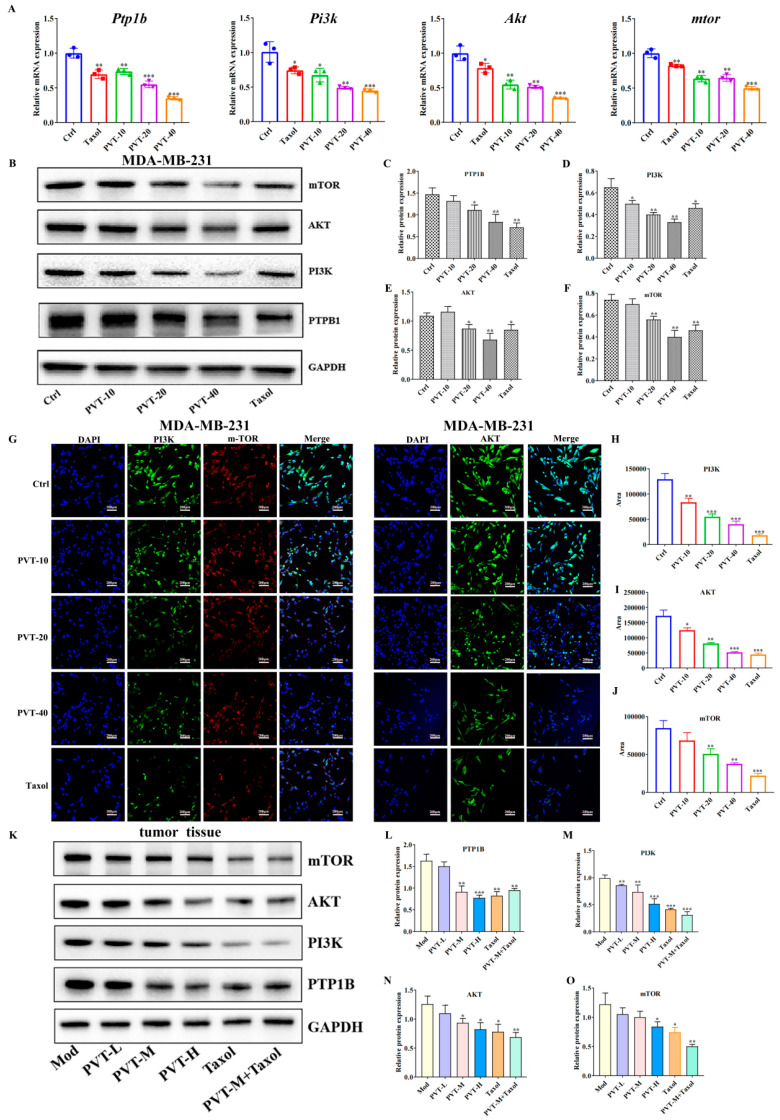
Effects of PVT on PTP1B/PI3K/AKT/mTOR pathway in MD-MB-231 cells and tumor tissues (mean ± SD, *n* = 3; * *p* < 0.05, ** *p* < 0.01, *** *p* < 0.001). (**A**) mRNA levels of *Ptp1b*, *Pi3k*, *Akt* and *mtor* in different groups of MDA-MB-231 cells; (**B**–**F**) expression levels of PTP1B, PI3K, AKT and mTOR in MDA-MB-231 cell lines were analyzed by Western blot after 48 h of PVT treatments; (**G**–**J**) expression levels of PI3K, AKT and mTOR in MDA-MB-231 cell lines were analyzed by immunofluorescence after 48 h of PVT treatments; (**K**–**O**) expression levels of PTP1B, PI3K, AKT and mTOR in tumor tissues of nude mice were analyzed by Western blot after 48 h of PVT treatments.

**Figure 10 ijms-26-01959-f010:**
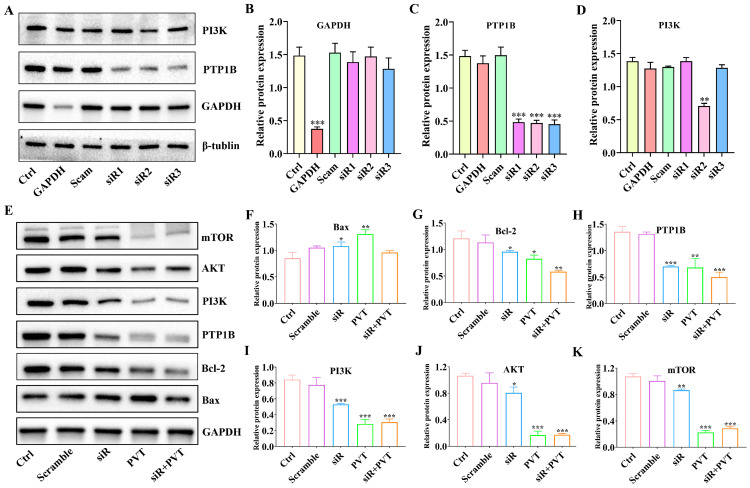
Regulation of PTP1B on PI3K/AKT/mTOR (mean ± SD, *n* = 3; * *p* < 0.05, ** *p* < 0.01, *** *p* < 0.001). (**A**–**D**) PTP1B was inhibited by siRNA; (**B**–**K**) expression of PI3K, AKT, mTOR, Bax and Bcl-2 proteins after silencing PTP1B.

**Figure 11 ijms-26-01959-f011:**
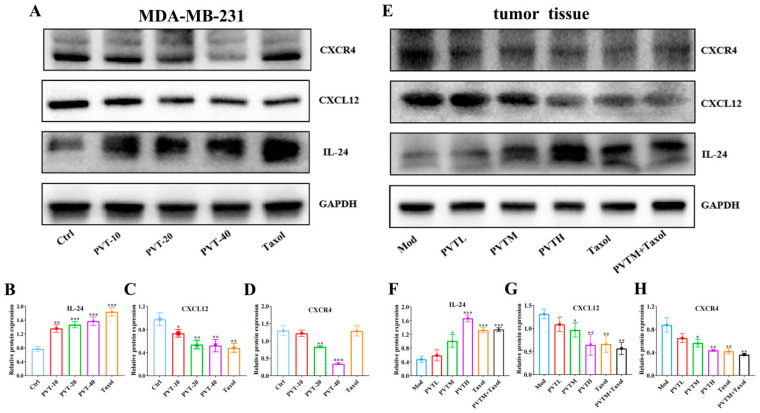
Effects of PVT on IL-24/CXCL12/CXCR4 pathway in MD-MB-231 cells and tumor tissues (mean ± SD, *n* = 3; * *p* < 0.05, ** *p* < 0.01, *** *p* < 0.001). (**A**–**D**) Expression of IL-24, CXCL12 and CXCR4 proteins in MDA-MB-231 cells; (**E**–**H**) expression of IL-24, CXCL12 and CXCR4 proteins in TNBC tumor tissues.

**Figure 12 ijms-26-01959-f012:**
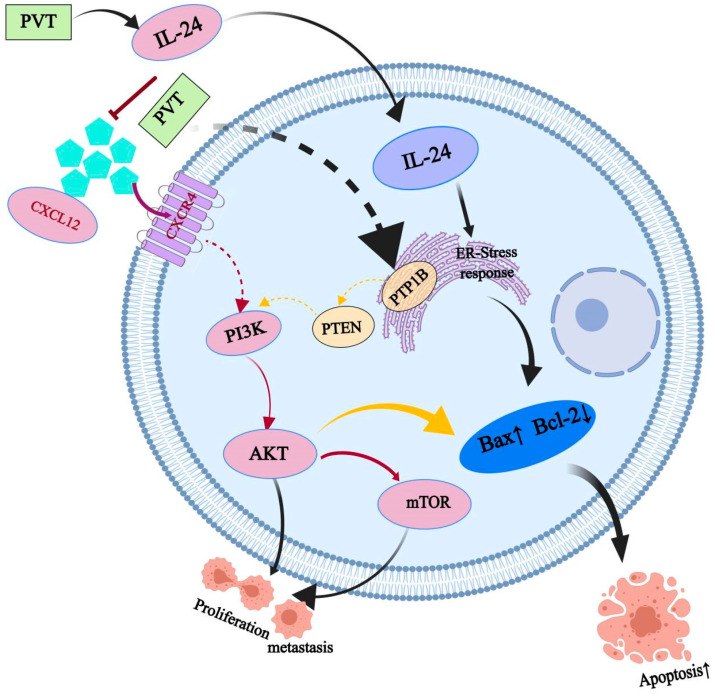
PVT inhibited tumor progression by regulating apoptosis and metastasis via the regulation of PTP1B/PI3K/AKT/mTOR and the IL-24/CXCL12/CXCR4 pathway in triple-negative breast cancer.

**Table 1 ijms-26-01959-t001:** Nineteen compounds of PVT were identified by LC-MS/MS.

Serial Number	*t*_R_/min	Measured Value [M-H]^−^	Theoretical Value	Molecular Formula	Compound	Characteristic Ion
1	14.066	487.3436	487.3430	C_30_ H_48_O_5_	Ananasic acid	487.3436
2	14.498	311.1689	311.1798	C_20_H_24_O_3_	Triptophenolide	311.1689, 271.1335, 225.1273
3	14.950	491.2732	491.2657	C_28_H_36_N_4_O_4_	Mucronine B	491.2659
4	15.021	285.0409	285.0408	C_15_H_10_O_6_	Kaempferol	285.0408
5(a)	15.755	469.3328	469.3333	C_30_H_46_O_4_	18α-Glycyrrhetinic acid	469.3328, 373.1238, 175.6028
6(b)	16.415	471.3476	471.3483	C_30_H_48_O_4_	Maslinic acid	471.3476, 409.3477
7(c)	16.773	471.3479	471.3486	C_30_H_48_O_4_	Hederagenin	471.3485
8(d)	18.843	617.3851	617.3849	C_39_H_54_O_6_	cis-p-Coumaroylcorosolic acid	617.3851
9(e)	19.152	499.3799	499.3804	C_32_H_52_O_4_	Oleananoic acid acetate	499.3799, 409.364
10(f)	19.527	455.3544	455.3546	C_30_H_48_O_3_	Betulinic acid	455.3544
11(g)	20.213	455.3534	455.3539	C_30_H_48_O_3_	Oleanolic acid	455.3534
12(h)	20.551	455.3531	455.3536	C_30_H_48_O_3_	Ursolic acid	455.3531
13	20.745	592.2687	592.7000	C_34_H_40_O_9_	Isomoreollic acid	592.2687, 515.2454
14(i)	21.650	453.3368	453.3382	C_30_H_46_O_3_	Liquidambaric acid	453.3366
15	26.171	381.3372	381.3378	C_24_H_46_O_3_	3-oxo-tetracosanoic acid	381.3372
16	27.313	714.5094	714.5074	C_39_H_74_NO_8_P	1-Palmitoyl-2-linoleoyl PE	714.5094
17	28.275	633.4378	633.4237	C_35_H_58_O_6_	Stigmasteryl glucoside	633.4378
18	33.054	293.1765	293.1837	C_17_H_26_O_4_	Embelin	293.1765
19	33.081	809.5176	809.5159	C_41_H_79_O_13_P	PI(16:0/16:0)	809.5159

**Table 2 ijms-26-01959-t002:** Information about triterpenoids in PVT.

Serial Number	Molecular Formula	Compound	Canonical SMILES
M1	C_30_H_48_O_4_	Maslinic acid	CC1(CCC2(CCC3(C(=CCC4C3(CCC5C4(CC(C(C5(C)C)O)O)C)C)C2C1)C)C(=O)O)C
M2	C_30_H_48_O_4_	Hederagenin	CC1(CCC2(CCC3(C(=CCC4C3(CCC5C4(CCC(C5(C)CO)O)C)C)C2C1)C)C(=O)O)C
M3	C_39_H_54_O_6_	cis-p-Coumaroylcorosolic acid	CC1CCC2(CCC3(C(=CCC4C3(CCC5C4(CC(C(C5(C)C)OC(=O)/C=C/C6=CC=C(C=C6)O)O)C)C)C2C1C)C)C(=O)O
M4	C_32_H_52_O_4_	Oleananoic acid acetate	CC(=O)OC1CCC2(C3CCC4C5CC(CCC5(CCC4(C3(CCC2C1(C)C)C)C)C(=O)O)(C)C)C
M5	C_30_H_46_O_4_	18α-Glycyrrhetinic acid	CC1(C2CCC3(C(C2(CCC1O)C)C(=O)C=C4C3(CCC5(C4CC(CC5)(C)C(=O)O)C)C)C)C
M6	C_30_H_48_O_3_	Betulinic acid	CC(=C)C1CCC2(C1C3CCC4C5(CCC(C(C5CCC4(C3(CC2)C)C)(C)C)O)C)C(=O)O
M7	C_30_H_48_O_3_	Oleanolic acid	CC1(CCC2(CCC3(C(=CCC4C3(CCC5C4(CCC(C5(C)C)O)C)C)C2C1)C)C(=O)O)C
M8	C_30_H_48_O_3_	Ursolic acid	CC1CCC2(CCC3(C(=CCC4C3(CCC5C4(CCC(C5(C)C)O)C)C)C2C1C)C)C(=O)O
M9	C_30_H_46_O_3_	Liquidambaric acid	CC(=C)C1CCC2(C1C3CCC4C5(CCC(=O)C(C5CCC4(C3(CC2)C)C)(C)C)C)C(=O)O

**Table 3 ijms-26-01959-t003:** The top five upregulated and downregulated genes in MDA-MB-231 cells after PVT intervention.

Gene ID	Gene Symbol	log2 (PVT/Ctrl)	Qvalue (PVT/Ctrl)	Gene ID	Gene Symbol	log2 (PVT/Ctrl)	Qvalue (PVT/Ctrl)
8549	*Lgr5*	5.2587	0.0035	5104	*Serpina5*	−1.9025	8.97 × 10^−5^
183	*Agt*	3.4494	5.20 × 10^−5^	3371	*Tnc*	−1.8073	2.69 × 10^−120^
440533	*Psg8*	3.2907	0.0159	3627	*Cxcl10*	−1.7788	0.0335
11009	*Il24*	2.9291	7.06 × 10^−7^	84443	*Frmpd3*	−1.7137	0.0217
375616	*Kcp*	2.8352	3.48 × 10^−13^	54492	*Neurl1b*	−1.5728	4.50 × 10^−42^

**Table 4 ijms-26-01959-t004:** Sequences of qRT-PCR primers.

Gene	Primer Sequences (5′−3′) FORWARD	Primer Sequences (5′−3′) REVERSE
*Gapdh*	TGACATCAAGAAGGTGGTGAAGCAG	GTGTCGCTGTTGAAGTCAGAGGAG
*Lgr5*	TAATCAGCTAAGACACGTACCC	GGGATTTCTGTTAACGCATTGT
*Agt*	GAGAAGATTGACAGGTTCATGC	GAAGTGGACGTAGGTGTTGAAA
*Psg8*	TGTGAAATACGGAACCCAGTGAGTG	TTGTTGATGGTGATGTAGGGCTTGG
*Il24*	TTTGTTCTCATCGTGTCACAAC	GTTTGAATGCTCTCCGGAATAG
*Kcp*	ACTGTGACATGCTCCTTGGTTGAC	GCCGTCCACAAACACCTCTTCC
*Serpina5*	CCAGAAAAGCTCAGAGAAGGAG	CAGAGTCCCTAAAGTTGGTAGG
*Tnc*	TAGTGAAAAACAATACCCGGGG	TGACATCTTTCACCTCGATCTG
*Cxcl10*	CTCTCTCTAGAACTGTACGCTG	ATTCAGACATCTCTTCTCACCC
*Frmpd3*	CACAATCACTCCTGAATCATCG	CGACATCATCTCTGACATGTCT
*Neurl1b*	TATGACCTTCAGTGTCAACCAG	GTGTAGATGACCGTGTCCAC
*Ptp1b*	CCAGCCAAAGGGGAGCCGTC	CTATGTGTTGCTGTTGAACA
*Pi3k*	GGAAGCAGCAACCGAAACAAAG	TCCACCACTACAGAGCAGGCATAG
*Akt*	GCAGGATGTGGACCAACGTGAG	GCAGGCAGCGGATGATGAAGG
*mtor*	CAACCAGCCAATCATTCGCATTCAG	ATGTCCGTTGCTGCCCATAAGTG

## Data Availability

The original contributions presented in this study are included in the article. Further inquiries can be directed to the corresponding author(s).

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
