# Peer review of "Triterpenes of Prunella vulgaris Inhibit Triple-Negative Breast Cancer by Regulating PTP1B/PI3K/AKT/mTOR and IL-24/CXCL12/CXCR4 Pathways"

_ijms, 2025, doi:10.3390/ijms26051959_

Round 1

Reviewer 1 Report

Comments and Suggestions for Authors

The manuscript entitled "Triterpenes of Prunella vulgaris inhibit triple-negative breast  cancer by regulating PTP1B/PI3K/AKT/mTOR and IL-2 24/CXCL12/CXCR4 pathway" was reviewed, and appropriate comments were indicated.

The subject is dealt with an important issue that now represent a hazardous growing outbreak, BC and its sub-type TNBC.  Authors explored the inhibitory effects of PVT on TNBC and 21 associated mechanisms with satisfied experiments and clear presentation with tables and figures.

The PVT could be further purified into separate compounds to explore the effect of individual compounds on TNBC to explain the role accurately.

Overall, this is a well-structured and informative article that successfully unravelling that PVT plays an anti-TNBC role by regulating the 30 PTP1B/PI3K/AKT/mTOR signaling pathway and the IL-24/CXCL12/CXCR4 signaling axis.  Apart from these suggested improvements, the content aligns well with the article's title and objectives, making it a robust and useful piece of scientific literature.

Thus, it could be suitable for publication after some modifications. Here are some suggestions/revisions:

Major issues:

Line 36: Revise “Labiatae” to “Lamiaceae”.

Line 38: Prunella vulgaris L.” addressed fully in Line 36. The following addressing thoughout the manuscript should be Prunella vulgaris or P. vulgaris.

Line 60: State the complete meaning of the abbreviation “TCM” as Traditional Chinese Medicine for the first time they appear in the text and then use them as abbreviated.

If possible, the resolution of charts within figures needs to be increased to valorize the valuability of this work.

Minor issues:

Lines 24 and 41: Avoid starting sentences with abbreviations as PVT and TNBC and so on.

Lines 37 and 41: Revise “meridians[1].” and “cancer[2].” make a space between text and reference’s brackets and so on.

Line 36: Revise “L.” should not be italic.

Lines 46 and 47: Revise “cancerand” and “threatensthe” make a space and so on.

Line 68: Revise “inflammation”

Line 68: Revise “Investigate”.

Author Response

Comment: Major issues:

Comment 1-1. Line 36: Revise “Labiatae” to “Lamiaceae”.

Response: Thank you for your comments. The issue has been revised in line 36 of the manuscript.

Comment 1-2. Line 38: “Prunella vulgaris L.” addressed fully in Line 36. The following addressing thoughout the manuscript should be Prunella vulgaris or P. vulgaris.

Response: Thank you for your comments. The following addressing of “Prunella vulgaris L.” thoughout the manuscript has been replaced with “Prunella vulgaris”.

Comment 1-3. Line 60: State the complete meaning of the abbreviation “TCM” as Traditional Chinese Medicine for the first time they appear in the text and then use them as abbreviated.

Response: Thank you for your comments. The complete meaning of “TCM” has been supplemented in line 61 of the manuscript, and then “Traditional Chinese Medicine” is abbreviated as “TCM”.

Comment 1-4. If possible, the resolution of charts within figures needs to be increased to valorize the valuability of this work.

Response: Thank you for your comments. The issue has been revised of the manuscript. Due to the size limitation of the uploaded manuscripts, the resolution of the pictures was reduced. We have re-uploaded the high-resolution pictures in the

manuscripts.

Comment: Minor issues:

Comment 2-1. Lines 24 and 41: Avoid starting sentences with abbreviations as PVT and TNBC and so on.

Response: Thank you for your comments. The issue has been revised in lines 23, 56 and so on of the manuscript. 

Comment 2-2. Lines 37 and 41: Revise “meridians[1].” and “cancer[2].” make a space between text and reference’s brackets and so on.

Response: Thank you for your comments. A space between text and reference’s brackets has been made of the manuscript. 

2-3. Line 36: Revise “L.” should not be italic.

Response: Thank you for your comments. The issue has been revised in line 36, of the manuscript.  

2-4. Lines 46 and 47: Revise “cancerand” and “threatensthe” make a space and so on.

Response: Thank you for your comments. The issue has been revised in lines 23, 45, 46 , 47, 50, 57, 82, 84, 87,91, 100, 105, 106, 117, 119, 173, 178, 212, 255, 272, 423, 426, 428, 438, 459, 460, 477, 478, 507, 529, 533, 536, 573, 593, 609, 610, 621,

622, 626, 638, 639, 642 and 643 of the manuscript.  

2-5. Line 68: Revise “inflammation”

Response: Thank you for your comments. The issue has been revised in line 69 of the manuscript, inflammation has been replaced by anti-inflammation.

2-6. Line 68: Revise “Investigate”.

Response: Thank you for your comments. The first letter of “Investigate” in line  91 of the manuscript should be in lowercase. The issue has been revised.

Reviewer 2 Report

Comments and Suggestions for Authors

The primary goal of the research presented here is to demonstrate pharmaceutical potential of an alcoholic extract from Prunella vulgaris L., a traditional medical herb in China. The authors conducted very comprehensive studies including chemical extraction of natural products, chemical analysis of the extract, and therapeutic evaluation of the extract both in vitro and in vivo. Subsequently, various molecular biology approaches were utilized to understand the mechanisms of action of the extract. Because therapeutic components are present as a mixture, precise elucidation of drug mechanism is (almost) impossible. Through multiple studies in molecular pharmacology both at the cell and animal levels, the authors were able to demonstrate that the P1B/PI3K/AKT/mTOR and IL-2 24/CXCL12/CXCR4 pathways were involved in the therapeutic efficacy of the extract for treating triple negative breast cancers, at least in part. The conclusion from the results is solid.  The manuscript is organized well and easy to follow. Some specific comments are listed below.

Major comment:

The current content in the section of “3. Discussion” mainly reiterates or summarizes the key findings in the second section of “Results”. I would suggest the authors focus on data interpretation, especially on something unexpected. Just to name a few, for example, (1) in Figure 5, Taxol was more effective in inducing apoptosis compared to PVT (Figures 5A-5F), but this could not be well explained by Bax upregulation and Bcl -2 downregulation (Figures 5G-5J);  (2) in Figure 8H, why would the combination of PVT-M and Taxol have lower expression of Bax than Taxol alone; and (3) in the xenograft model, the combination of PVT-M and Taxol was at least equally effective compared to PVT-H (Figure 7B), but PVT-H seemed to be more effective on regulating Bax/Bcl-2.  All these differences may be attributed to the fact that the extract is a mixture, but it would be appreciated if the authors could attempt to provide some insights based on literature and/or current knowledge.

Minor points:

  1. In line 30, “shown” should change to “showed”.
  2. In lines 60 and 128, define the abbreviations “TCM” and “PPI”, respectively.
  3. In the legend of several Figures where Taxol is used, define its dosage.
  4. Would “Table 4” in line 215 be “Table 3”?
  5. Would the column log2 (PVT/Ctrl) in Table 4/3 indicate up- or down-regulation? If so, would Frmpd3 be downregulated, opposite to Il24? This would be different from the statement in line 216.
  6. Based on Figures 10C and 10D, siR2 seems not specific for knocking down PTP1B compared to the other two siRNAs. Why would the authors choose siR2?  Or would the authors aim to knock down PTP1B and PI3K at the same time?
  7. There is no need to define TNBC in both lines 338 and 347.
  8. In section 4.6, could the authors provide more detailed information on how scratch area was calculated in the migration assay? Is any software involved?
  9. In line 510, “and” between “was” and “resuspended” should be removed.
  10. In line 532, “Table 1” should change to “Table 4”.
  11. From line 549 to line 551, the information on dosage seems incomplete.
  12. In line 571, would “hydrated” be “dehydrated”?
Comments on the Quality of English Language

Some proofreading in English would improve readability, but extensive editorial service may not be necessary.

Author Response

1. Major comment:

The current content in the section of “3. Discussion” mainly reiterates or summarizes the key findings in the second section of “Results”. I would suggest the authors focus on data interpretation, especially on something unexpected. Just to name a few, for example, (1) in Figure 5, Taxol was more effective in inducing apoptosis compared to PVT (Figures 5A-5F), but this could not be well explained by Bax upregulation and Bcl -2 downregulation (Figures 5G-5J);  (2) in Figure 8H, why would the combination of PVT-M and Taxol have lower expression of Bax than Taxol alone; and (3) in the xenograft model, the combination of PVT-M and Taxol was at least equally effective compared to PVT-H (Figure 7B), but PVT-H seemed to be more effective on regulating Bax/Bcl-2.  All these differences may be attributed to the fact that the extract is a mixture, but it would be appreciated if the authors could attempt to provide some insights based on literature and/or current knowledge.

Response: Thank you for your comments. We have made careful revisions to the discussion section. We have analyzed and discussed some unexpected situations that emerged from the research results and illustrated them by citing references. The

specific revision details have been presented in the manuscript.

2. Minor points:

Comment 2-1. In line 30, “shown” should change to “showed”.

Response: Thank you for your comments. The issue has been revised in line 30  of the manuscript.

Comment 2-2. In lines 60 and 128, define the abbreviations “TCM” and “PPI”, respectively.

Response: Thank you for your comments. The issue has been revised in lines 61 and 133 of the manuscript. The definition of “TCM” is “Traditional Chinese Medicine”. The definition of “PPI” is “Protein-Protein Interaction Networks”.

Comment 2-3. In the legend of several Figures where Taxol is used, define its dosage.

Response: Thank you for your comments. The dosage and concentration of Taxol had been defined in the section 4 Materials and methods of the manuscript.

Comment 2-4. Would “Table 4” in line 215 be “Table 3”?

Response: Thank you for your comments. The issue has been revised in line 225 of the manuscript. The “Table 4” in line 225 is “Table 3”.

Comment 2-5. Would the column log2 (PVT/Ctrl) in Table 4/3 indicate up- or down-regulation? If so, would Frmpd3 be downregulated, opposite to Il24? This would be different from the statement in line 216.

Response: Thank you for your comments. Upon qRT-PCR verification, it was found that PVT could up-regulate Il-24 and down-regulate Serpina5, Tnc, Cxcl10 and Neurl1b, but up-regulate Frmpd3 and down-regulate Lgr5 and Psg8, which was contrary to the transcriptomic results. Generally speaking, RNA-seq is used for large-scale screening to reflect the overall trend of gene expression changes in samples, but it cannot guarantee that the trend of every gene's expression change is

consistent with that of qPCR.

Comment 2-6. Based on Figures 10C and 10D, siR2 seems not specific for knocking down PTP1B compared to the other two siRNAs. Why would the authors choose siR2?  Or would the authors aim to knock down PTP1B and PI3K at the same time? 

Response: Thank you for your comments. This is because downstream PI3K is also taken into account, as PI3K is also regulated by other upstream factors. So I chose siR2. Since the drug used in this study is extracts of traditional Chinese medicine, not a single component, and may act through multiple pathways, it was found in the enrichment of network pharmacological pathways in the early stage that PVT can act through the PI3K-Akt signaling pathway, so the role of siRNA on PI3K can be considered here.

Comment 2-7. There is no need to define TNBC in both lines 338 and 347.

Response: Thank you for your comments. The issue has been revised in lines 348 and 353 of the manuscript. 

Comment 2-8. In section 4.6, could the authors provide more detailed information on how scratch area was calculated in the migration assay? Is any software involved?

Response: Thank you for your comments. The scratch area was calculated in the migration assay via ImageJ software and the calculation formula has been supplemented..

Comment 2-9. In line 510, “and” between “was” and “resuspended” should be removed.

Response: Thank you for your comments. The issue has been revised in line 544 of the manuscript. 

Comment 2-10. In line 532, “Table 1” should change to “Table 4”.

Response: Thank you for your comments. The issue has been revised in line 565 of the manuscript. In line 565, “Table 1” has been changed to “Table 4”.

Comment 2-11. From line 549 to line 551, the information on dosage seems incomplete.

Response: Thank you for your comments. The issue has been revised from line 583 to 587 of the manuscript. The information on dosage has been completed.

Comment 2-12. In line 571, would “hydrated” be “dehydrated”?

Response: Thank you for your comments. The issue has been revised in line 607 of the manuscript. 
